# Health Recommender Systems Development, Usage, and Evaluation from 2010 to 2022: A Scoping Review

**DOI:** 10.3390/ijerph192215115

**Published:** 2022-11-16

**Authors:** Yao Cai, Fei Yu, Manish Kumar, Roderick Gladney, Javed Mostafa

**Affiliations:** 1School of Information and Library Science, The University of North Carolina at Chapel Hill, Chapel Hill, NC 27599, USA; 2School of Economics and Management, Nanjing University of Science and Technology, Nanjing 210094, China; 3Carolina Health Informatics Program, The University of North Carolina at Chapel Hill, Chapel Hill, NC 27599, USA; 4Public Health Leadership Program, The University of North Carolina at Chapel Hill, Chapel Hill, NC 27599, USA

**Keywords:** health recommender system, research area, user model, recommendation technology, system evaluation

## Abstract

A health recommender system (HRS) provides a user with personalized medical information based on the user’s health profile. This scoping review aims to identify and summarize the HRS development in the most recent decade by focusing on five key aspects: health domain, user, recommended item, recommendation technology, and system evaluation. We searched PubMed, ACM Digital Library, IEEE Xplore, Web of Science, and Scopus databases for English literature published between 2010 and 2022. Our study selection and data extraction followed the Preferred Reporting Items for Systematic Reviews and Meta-Analyses extension for Scoping Reviews. The following are the primary results: sixty-three studies met the eligibility criteria and were included in the data analysis. These studies involved twenty-four health domains, with both patients and the general public as target users and ten major recommended items. The most adopted algorithm of recommendation technologies was the knowledge-based approach. In addition, fifty-nine studies reported system evaluations, in which two types of evaluation methods and three categories of metrics were applied. However, despite existing research progress on HRSs, the health domains, recommended items, and sample size of system evaluation have been limited. In the future, HRS research shall focus on dynamic user modelling, utilizing open-source knowledge bases, and evaluating the efficacy of HRSs using a large sample size. In conclusion, this study summarized the research activities and evidence pertinent to HRSs in the most recent ten years and identified gaps in the existing research landscape. Further work shall address the gaps and continue improving the performance of HRSs to empower users in terms of healthcare decision making and self-management.

## 1. Introduction

People are increasingly using the internet to search for health information. Google receives more than 1 billion health questions every day and 7% of Google’s daily searches are health related (Beckers health https://www.beckershospitalreview.com/healthcare-information-technology/google-receives-more-than-1-billion-health-questions-every-day.html, accessed on 6 February 2021). Pew Research Center even showed 8 in 10 internet users have searched for health information (e.g., diet, fitness, drugs, health insurance, treatments, doctors, and hospitals) (Pew Research Center https://www.pewresearch.org/internet/2005/05/17/health-information-online/, accessed on 6 February 2021). Individuals are more engaged in healthcare decision-making and self-management of disease when they have access to online health information resources [1].

However, the volume and complexity of health information can easily exceed an individual’s information processing capacity and lead to information overload [2], resulting in incomplete or inaccurate information consumption. In addition, the online information is usually not tailored to each patient’s specific needs [3]. Furthermore, users’ health literacy levels vary, and some lack the adequate skills to understand the medical terminology and vocabulary, or evaluate the actual relevance of the extracted data, or check the validity of the information sources [4]. Overall, it can be a challenging experience for consumers seeking health information on the internet [5,6].

Recommendation systems can help people avoid information overload by adopting computer-based intelligent techniques. These systems typically provide users with a list of recommended items that they deem as relevant or enable users to evaluate the relevancy and ease the task of information searching.

A health recommendation system (HRS) is a specialized recommender system that supplies a user with personalized health information, which is meant to be highly relevant to the user’s health profile [7]. In addition, an HRS is capable of automatically identifying and recommending appropriate materials (such as recommending diagnoses, health insurance, clinical pathway-based treatment methods, and alternative medicines) to users based on their specific health conditions and needs. Therefore, an HRS can empower people with relevant health information, enable them to have more control on their own health management, and engage them in important health decision-making [8]. HRS introduces an opportunity for the health care industry to transition from a traditional to a more personalized paradigm in a telehealth environment.

Given the increasing significance of HRS and its potential impact on patients’ decision-making and self-management, a comprehensive scoping review of the relevant literature related to this topic is warranted, describing the HRS research landscape and identifying potential research gaps for further exploration.

In recommender systems, three main entities play crucial roles, namely user, recommended item, and recommendation methods [9,10,11]. Based on the users’ preferences and their information needs, a recommender system will generate a list of items based on its recommendation methods and present them to users. Based on the three main entities and constructs of HRSs in previous studies [12,13,14,15,16,17,18,19], we conclude five major aspects concerning an HRS: health domain, user, recommended item, recommender system, and system evaluation (Table 1).

To map the existing research literature pertaining to the use of HRS for patients’ decision-making and self-management, we conducted a scoping review to identify and analyze existing recommender systems in the health domain by following the protocol of Preferred Reporting Items for Systematic Reviews and Meta-Analyses extension for Scoping Reviews (PRISMA-ScR). Our research questions (RQ) are below:

RQ1—what are the health research domains covered by HRS?

User model is the direct source for a system to know the users’ preferences and information needs. When there is little or no information about a particular user, it becomes almost impossible for the recommender system to propose a solution. In order to generate personalized information, recommender systems need to design a comprehensive set of data about the user. User modeling is the process of determining which user characteristics are relevant to a service and designing a structure that will capture these attributes.

RQ2—what are the target user of HRS? And what user attributes are used to build user model?

The HRS should be trustworthy and reliable so that end patients can use this system to their benefit. 

RQ3—what are the recommended items? And what is the information source of HRS are.

Recommendation algorithms is an important part of recommendation system. It aimed at suggesting relevant items to users.

RQ4—what are the recommendation technologies/filtering approaches used to generate recommendations?

For the success of the recommender system, it is important to choose what type of criteria are used to evaluate the recommender system.

RQ5—How do the existing HRSs evaluate their performance?

We identified a number of key principles and processes with respect to the above five research questions for effectively designing future HRS to ensure usability and reliability.

## 2. Methods and Materials

### 2.1. Literature Search 

We developed a comprehensive search strategy by including key concepts (e.g., recommender system, health, personalization, etc.) and their synonyms (Appendix A). The search strategy was executed in PubMed, ACM Digital Library, IEEE Xplore, Web of Science, and Scopus. Among these key databases, PubMed, ACM digital library, and IEEE Xplore are the most commonly used bibliographic databases in most HRS review papers [14,20]. In addition, we included two widely used bibliographic databases—Web of Science and Scopus [13]. The literature search was conducted on 20 November 2020 and retrieved articles were limited to publications between 2010 and 2022.

### 2.2. Inclusion and Exclusion Criteria

Studies were eligible for inclusion in this review if all the following criteria were met: (1) information recommendation was related to human health, and the primary focus was to promote health [20]; (2) the study applied personalization technique (e.g., for a recommender system, recommendation to be carried out in a personalized manner and they must be based on information related to the user that must be reflected in the user profile or user model) [11]; (3) information recommendation was generated automatically and provided through the usage of technology. The intervention where personalization was provided manually was excluded, as the user model for delivering the personalization cannot be identified in a manual process [13]; (4) a user model was generated for providing recommendations; (5) the study was published in English and between 2010 and 2022; (6) the study was published in peer-reviewed journals or conference proceedings; and (7) target users of the recommender system were patients or the general public. (e.g., ref. [21] aims to help doctors personalize the prescribed medications was excluded).

The exclusion criteria were (1) information recommendation was not related to human health; (2) personalization was only at the level of using the user’s name on the displaying page; (3) recommendation was generated manually, even though it was delivered through technology (e.g., via a website); (4) no user model was disclosed for generating recommendations; (5) the target users of recommender systems were healthcare professionals only (e.g., physicians, nurses, and other practitioners who are directly responsible for patient care); (6) gender- or culture- based general recommendations; (7) studies were not in English language, or the full-text articles were not accessible through the institutional interlibrary loan service.

### 2.3. Study Selection

We followed the Preferred Reporting Items for Systematic Reviews and Meta-Analyses extension for Scoping Reviews (PRISMA-ScR) for study selection and data extraction [8]. The PRISMA for scoping reviews was published in 2018. The checklist contains 20 essential reporting items and 2 optional items to include when completing a scoping review. Detailed items are included in Appendix A. All retrieved citations were aggregated, de-duplicated, and screened at title-abstract and full-text levels using Mendeley Reference Manager [22].

The first phase involved title, abstract, and keyword review as obtained from the database search. This phase was applied to all aggregated and deduped results from the databases search. The second phase included reviewing the full text of the articles. This was done by obtaining PDF documents for each of the articles that met the inclusion criteria. The full texts were analyzed using the inclusion and exclusion criteria, and the studies that passed the full-text screening were included in the scoping review. The screening procedure and study selection was undertaken by Y.C. and two graduate research assistants and then independently verified by M.K. and R.G. Any disagreements were resolved by discussion and consultation with a third researcher before reaching consensus.

## 3. Results

### 3.1. Screening and Study Selection

Of 20,255 articles retrieved from literature search, 63 were included for data extraction and analysis (Figure 1). Complete descriptions of included publications are in Appendix A.

### 3.2. RQ1 Health Domain

There were 24 health domains addressed by the HRSs in 63 studies (Table 2).

### 3.3. RQ2 User


Target user


The target users of an HRS included both the general public and patients who have a specific disease or health symptom (Figure 2).
User model

Each of the 63 studies created a different user model and defined their users through various attributes (Table 3). Generally, the user attributes in the included studies were distributed across 6 categories: demographics, medical condition, food intake, physical activity, explicit data, and implicit data (Appendix A).

### 3.4. RQ3 Recommended Item


Recommended information


Generally, there are 10 types of recommended health information, and their distribution is shown in Table 4.
Information source

The included studies utilized the following 5 types of information sources to generate their recommended information (Appendix A).
(1)Credible and authoritative websites, guidelines, and books: nine studies [28,33,39,51,56,62,78,84,85] recommended information content from websites approved by healthcare professional societies such as “American Cancer Society.” Another seven studies [31,35,39,51,61,80,81] recommended content from guidelines published by professional associations such as the American College of Sports Medicine Training Progression guidelines. Seven studies [25,32,34,39,46,84,85] recommended content from books (e.g., Weight-Less book [25]).(2)Domain experts: in 16 studies [23,30,35,36,44,45,54,57,59,60,62,63,64,65,83,84] the information source was created by health sciences researchers or clinical professionals (e.g., researchers specializing in behavioral change theories, smoking cessation psychologists, pulmonologists, or nutrition specialists).(3)Similar patients: five studies recommended items extracted from patient databases (e.g., a database of 132 obese youth who had succeeded in managing obesity [42]), social networks sites by crowdsourcing [67], or patients with similar health problems [31,59,79].(4)Government database: four studies used the United States Department of Agriculture (USDA) food composition guidelines [25,51,52,76]; other studies extracted dietary information from the Taiwan food nutrition database of the Food and Drug Administration [38], Taiwanese Snacks Nutrition Analysis [37], Food Composition Database (FCD)”, and Japan Preventive Association of Life-style related Disease (JPALD) [78].(5)Other resources: a range of other information resources were utilized for recommended items, including online health community [48,49,50], online registered doctors [66,68,70,71], restaurants [40], and psychotherapy approaches [47,74] downloaded from Google Play Store.

### 3.5. RQ4 Recommendation Technology


Algorithms


Six types of algorithm approaches were summarized below (Table 5) used for recommendation.
System name and user interface

Fifty-eight percent of the included studies (37/63) disclosed the names of their HRSs and, 19 of them were web-based systems [33,36,43,54,56,57,58,59,60,68,69,70,73,75,76,78,80,81,83], and 17 were mobile app systems [23,25,28,30,31,40,41,42,44,45,46,51,63,64,74,79,85].

### 3.6. RQ5 System Evaluation

Fifty-nine of the 63 studies reported evaluations of their HRSs, in which two types of system evaluation methods were applied. A few studies even used both types of methods. (1) System evaluation involving users: while there were 22 studies that recruited the end users of the systems and tested the feasibility of the HRSs [25,27,28,32,33,41,42,43,44,55,57,59,63,64,68,69,70,76,78,80,81,82], fourteen studies measured the performance of HRS through comparing the accuracy of different groups [26,29,31,36,42,45,48,49,53,60,61,66,67,74]. For example, an HRS was compared with health conditions in general or no recommendation scenarios. (2) System evaluation not involving users: 19 studies either directly invited experts to evaluate the system or compared HRS generated recommendations with expert recommendations for the same scenario [23,24,30,34,35,37,38,39,47,51,52,56,58,59,62,71,72,84,85].

A variety of metrics were applied to the performance assessment of personalization. (1) forty-six percent of the studies (29/63) used performance criteria (e.g., recall, mean reciprocal rank, normalized discounted cumulative gain, precision, mean absolute error, receiver operating characteristics, and F1) [23,29,34,35,36,37,38,39,41,42,45,47,48,49,50,51,52,58,59,62,63,64,66,68,72,74,75,80,81]. (2) twenty studies used user-centered usability criteria (e.g., usability, satisfaction, acceptance, perceived usefulness) [27,28,30,31,32,33,40,44,55,56,57,60,63,64,65,67,70,79,84,85]. (3) Ten studies used health related outcomes (e.g., balance of nutrition intake, behavior change in walking lengths or food intake calories, 30 days cessation, blood pressure control and reduce, weekly completed physical activity volume, and time to fall asleep) [25,26,53,60,61,63,64,65,77,79].

## 4. Discussion

Systematically identifying and analyzing 63 studies published between 2010 and 2022, this review added to existing HRS research literature in a couple of ways. First, it provided an overview of the recent research landscape regarding personalized health information recommendation of HRSs. Second, compared with previous review studies on HRSs [12,13,14,15,16,17,18,19] (Appendix A), this study is more comprehensive and rigorous in terms of literature search, study selection, and analysis of key constructs of HRSs by addressing the following identified research gaps. (1) Some reviews did not explore user models and recommendation algorithms, while we reviewed and summarized user modeling and algorithms that included studies used for item recommendation. (2) Reviews only focused on physical activity; (3) Reviews only investigated mobile based HRS; (4) Reviews did not explore evaluation methods, while we extracted and presented evaluation methods and metrics from each included study. Third, despite HRSs’ recent popularity and variety of application in numerous health promotion domains, HRSs are still in their infancy. Although some HRSs are already applied to healthcare, there is still a long way before they can be widely adopted in health-related domains. By summarizing existing research, we’ve identified several challenges in the design and development of existing HRSs, which can help make future HRSs more effective, robust, and accurate.

### 4.1. RQ1 Health Domain

There were 24 health domains in 63 studies. Chronical disease, healthy lifestyle, and health service are the three major health domains of the reviewed HRSs, accounting for 71% (45/63) of all included studies. This pattern resonates with the increasing patient engagement activities in self-management and medical decision-making [2]. However, the low presence of other health domains calls for HRSs targeting more diversified areas.

For example, chronic obstructive pulmonary disease (COPD), which includes emphysema and chronic bronchitis, makes breathing hard for the 16 million Americans who have been diagnosed with it. Millions more suffer from COPD but have not been diagnosed and are not being treated (https://www.cdc.gov/copd/features/copd-symptoms-diagnosis-treatment.html, accessed on 1 November 2022). An HRS can provide a personalized treatment program that teaches patients how to manage COPD symptoms and improve their quality of life. Plans may include learning to breathe better, how to conserve their energy, and recommended types of food and exercise. Another example is chronic constipation. The number of people affected by chronic constipation is 63 million (https://www.niddk.nih.gov/health-information/health-statistics/digestive-diseases, accessed on 1 November 2022). The symptoms usually can go away with personalized self-care.

The COVID-19 pandemic has exacerbated the constrained resources of our healthcare system [87]. HRSs can translate healthcare professional service into digital formats sopatients can access services and engage with interventions without requiring medical staff to be in-the-loop, greatly reducing the burden on the medical system. On the other hand, HRS makes the home-based isolation possible, greatly reducing disease spread and decreasing the risk of exposure by avoiding going to hospitals except in necessary conditions. Thus, disease specific HRSs are needed to address specific health topics and conditions and empower patients to manage symptoms and make personalized health decisions.

### 4.2. RQ2 User

Without proper profiling or obtaining user information, HRS is not able to provide personalized recommendations [88]. In order to generate personalized information, HRS need to collect user data. Thus, user modeling is the process of de-ermining which user characteristics are relevant to a service and then a system structure is designed accordingly to capture these attributes.

We found that the attributes of a user model in included studies are distributed across six categories: demographics, medical condition, food intake, physical activity, explicit, and implicit data. Questionnaires [28,30,31,39,46,57,59,60,63,64,65], electronic health records (EHRs) [39,59,64,85], and sensors [26,30,31,40,45,61,74,75,79,80,85] are the three major sources where HRSs extracted users’ health data. Particularly, EHRs contain not only medical treatment records but also patients’ health data, which makes it the most valuable source for HRSs to build accurate user models and make highly personalized recommendations. In addition, EHR data can be automatically exported to an HRS saving the manual data entry by patients. However, we only found 4 studies in which EHR data were utilized. Patient data privacy and legal barriers of data sharing may prohibit researchers from integrating EHRs with HRSs, which is a concerning issue for future research.

Patients’ health condition and preferences are changing over time. One unique requirement of the HRS would be to suggest health information in alignment with patients’ changing health status and updated knowledge of the disease. Therefore, user modelling needs to be adaptable to reflect the latest user needs, it must be dynamic. [88]. However, we found that only 7 studies [24,28,33,60,74,79,80] built dynamic user models. Among them, 2 studies [28,60] included a questionnaire that patients can take when their needs changed. Most HRSs examined in this study fall short of generating a dynamic user model and reflecting the updated patient health status. Thus, future studies shall aim at building dynamic user models by regularly and automatically importing patient health data and recommending personalized information relevant to the latest patients’ health conditions.

### 4.3. RQ3 Recommended Item

Two major types of recommended items were disclosed in this study. (1) Personalized disease-related information or patient educational material (14/63 studies), which is consistent with the findings from the Pew Internet and American Life Project. Providing patients with disease-specific information is an essential component of healthcare services because it equips patients with knowledge on symptoms, diagnostic tests, treatment options, side effects [89], and skills required for self-management and informed decision-making [90]. (2) Personalized dietary information (13/63 studies): a healthy diet helps to prevent noncommunicable diseases (e.g., diabetes, cardiovascular diseases, cancer, and other conditions linked to obesity) (World Health Organization https://www.who.int/health-topics/healthy-diet#tab=tab_1, accessed on 6 February 2021). An HRS can suggests food options that cater to individuals’ health goals and helps users develop healthy eating behaviors by following the recommendations. Searching for diet- and nutrition-related information is the second largest health information search behavior among Internet users,^2^ and it is also one of the most recommended items disclosed by this study.

Approximately one-third of Americans go online to research their health problems and they also use social networks to reach out to others with similar health conditions [91]. We identified “thread discussion” and “similar user” as recommended item in our results. However, we found the scope of recommended items was still very limited. Other important health topics such as health insurance, environmental health hazards, and immunizations were not covered in the examined studies at all.

### 4.4. RQ4 Recommendation System

Compared with the traditional collaborative filtering and content-based recommendation technique, the knowledge-based method is more appropriate in the context of an HRS, which was confirmed by the included studies. The knowledge-based method can personalize a user model to match the user characteristics through knowledge modelling (e.g., ontologies), alleviating some conventional drawbacks such as cold-start and rating sparsity problems. This study found most rules in knowledge-based system were built by a health domain expert. Future HRS design and development shall consider taking advantage of existing open-accessible knowledge bases (e.g., Wikipedia, DBpedia) in addition to health experts.

Among the included 63 studies, 36 HRSs were either web- or mobile-based and 17 were mobile-based-only applications. Since 81% adults own a smartphone (Statista, https://www.statista.com/statistics/219865/percentage-of-us-adults-who-own-a-smartphone/ accessed on 6 February 2021), mobile-based applications have great potential due to the convenience and benefits of reduced intervention implementation time, reaching broad and diverse populations, and greater accessibility, etc.

### 4.5. RQ5 System Evaluation

For the success of the recommender system, it is important to choose what type of criteria are used to evaluate the recommender system. Conventionally, recommender systems were evaluated based on criteria borrowed from information retrieval [92,93]. Common metrics used in the evaluation are: precision, recall, F-measure, ROC-Curve, and RMSE. In our reviewed papers, 29 included studies used those common metrics to assess the effectiveness of HRSs. The evaluation criteria of the recommender system are very necessary to measure the strength of an HRS based on user-centered usability criteria. Twenty studies used patient acceptance and satisfaction to evaluate the system. Besides, the measure of the effectiveness of a health recommender system depends on behavior evaluations. For example, when monitoring the health progress of patients and providing suggestions for treatment, keep track of activities. Those additional evaluation metrics included user-centered measurement and health-related outcomes, confirming that HRSs shall be measured by aspects beyond accuracy objectives [94]. However, no clinical trial was observed from the included studies and many studies involving users have a sample size of less than 30 participants [24,25,26,27,33,34,35,37,41,43,44,51,55,58,59,68,76,80,81]. Therefore, the clinical or health outcomes of the HRSs are not generalizable due to the limited sample size. In the future, clinical trials shall be considered to investigate the efficacy of HRSs.

### 4.6. Limitations

This study has a couple of limitations. First, we only reviewed studies published in the last ten years. The results and conclusion only apply to the examined time frame. Second, non-English literature was excluded from the review. Thus, we may omit the perspectives from the HRS studies published in other languages.

## 5. Conclusions

We summarized the highly influential works in health recommendation systems from the past 12 years regarding five key constructs (i.e., health domain, user, recommended item, recommendation technology, and system evaluation). Overall, at least 63 HRSs were developed to provide patients or general users with personalized recommendations on health information. These studies involved twenty-four health domains, both patients and the general public as target users, and ten major recommended items. The most adopted algorithm of recommendation technologies was the knowledge-based approach. In addition, fifty-nine studies reported system evaluations, in which two types of evaluation methods and three categories of metrics were applied. However, HRS developments are still in their infancy. This study identified several challenges in the design and development of existing HRSs, including lack of dynamic user modeling, limited scope of recommended items, and small sample size in system evaluation. The findings from this study shed light on future research direction, which shall focus on the identified research gaps and continuously improve HRS performance and usability to improve users’ health care and well-being.

## Figures and Tables

**Figure 1 ijerph-19-15115-f001:**
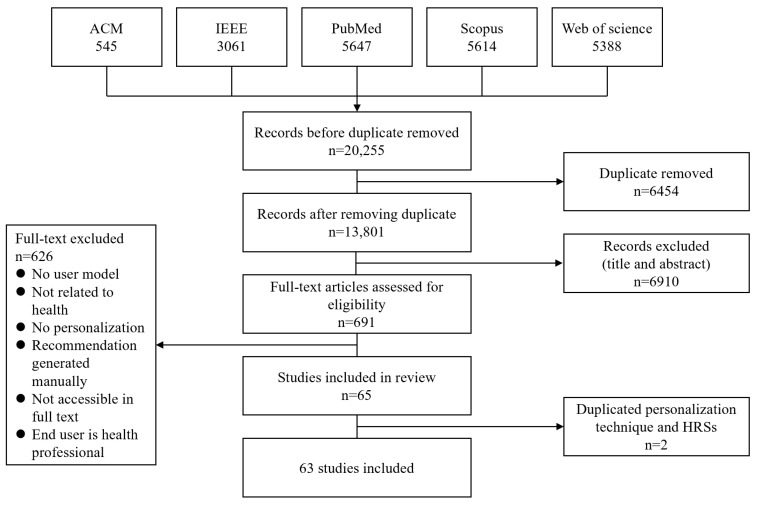
Study identification and selection.

**Figure 2 ijerph-19-15115-f002:**
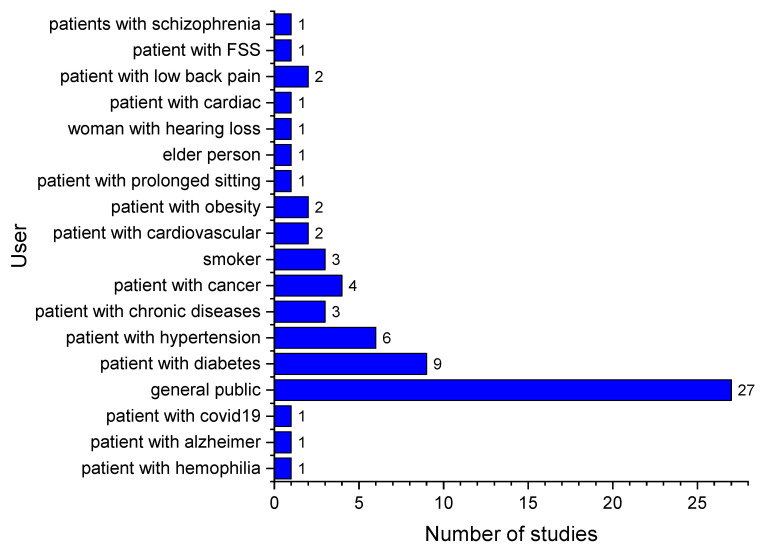
Distribution of target user.

**Table 1 ijerph-19-15115-t001:** Main components of HRS.

HRS Aspect		Description
Health domain		The involved specific health condition, issue, or disease
User	Target user	Targeted user population who receives health information recommendations
	User model	User-related attributes that are used for modeling and generating recommendations
Recommended item	Recommended information	The information that is being recommended by an HRS
	Information source	The information source where the recommended item comes from
Recommendation technology	Algorithms	Information filtering methods/techniques that are used to select recommended items
	System name and user interface	The name of an HRS and its user interface format (web vs. mobile)
System evaluation		Methods and metrics used to assess the performance and effectiveness of an HRS

**Table 2 ijerph-19-15115-t002:** The distribution of health domain.

Health Domain	Study	Health Domain	Study
Asthma	[23]	Hypertension	[24,25,26]
Cancer	[27,28,29,30]	Low back pain	[31,32]
Cardiovascular disease	[33]	Medicines and drugs	[34,35]
Chronic disease	[36,37,38,39]	Obesity	[40,41,42,43]
Depression	[44,45,46,47]	Online health forum	[48,49,50]
Diabetes	[43,51,52,53,54,55,56]	Sexual and reproductive disease	[57]
Elder care	[58]	Schizophrenia	[59]
Functional somatic symptoms (FSS)	[60]	Sleep problem	[61]
Heart disease	[62]	Smoking cessation	[63,64,65]
Health service	[66,67,68,69,70,71,72,73]	Stress	[74]
Healthy lifestyle	[75,76,77,78,79,80,81,82]	COVID-19	[83]
Alzheimer	[84]	Hemophilia	[85]

**Table 3 ijerph-19-15115-t003:** Attributes of user model.

User Attribute			Number of Studies	Study
Demographics		name, gender, age, body mass index, location, average charge, ethnicity, education level, marital status, blood type, weight, height, hip, waist, occupation, address, and income	34	[24,25,27,29,31,32,33,34,35,37,38,39,41,42,43,45,51,54,55,57,58,61,62,63,64,66,67,68,69,75,76,81,82,84]
Medical condition	Health status	symptoms, complications, treatments, health checks, diagnosis, diseases, severity, physiological, capability, depression, sleep, and pregnancy	49	[23,27,28,31,32,33,34,36,37,38,39,40,42,43,45,51,55,56,57,58,59,61,64,66,68,70,72,75,76,78,81,83,84,85]
	Laboratory test	blood glucose, total cholesterol, triglyceride, high/low density lipoprotein, uric acid, blood pressure, hypertrophy, carotid artery plaque, carotid femoral pulse wave velocity, ankle brachial index, and heart rate		[24,25,26,30,33,34,35,38,39,52,53,54,55,62,68,74]
	Disease history	family history, allergy, and smoking or alcohol history		[24,25,27,33,35,39,51,56,58,62,63,69,82]
	Medication	routine medicine that patient took		[34,39]
Food intake		food types, food size, food names and ingredients, and dietary record	15	[25,33,38,39,42,52,53,54,58,75,76,77,78,79,82]
Physical activity		activity level (high, moderate, low), type of activity, duration, past activity, intensity, and frequency	24	[26,27,30,33,37,38,39,41,43,45,53,54,55,56,58,60,61,75,79,80,81,82,85,86]
Explicit data (information preference provided by the user)		studies used the direct input preference from patients to generate recommendations OR utilized user ratings	31	[23,25,27,28,29,30,33,37,38,43,44,45,46,47,48,51,60,61,63,64,65,68,69,70,71,73,75,76,78,82,83]
Implicit data		generated from users’ interactions with the system (e.g., navigation or interaction logs)	13	[24,45,47,48,49,50,51,57,64,65,66,67,85]

**Table 4 ijerph-19-15115-t004:** The distribution of recommended information.

Recommended Item	Number of Studies	Study
Disease-related information or education material	14	[23,25,28,29,32,33,39,43,51,54,57,59,61,62,85]
Personalized diet information	13	[25,27,36,37,38,40,41,42,52,56,76,77,78]
Bolus insulin	2	[53,55]
Personalized physical activity	9	[24,26,30,31,44,45,60,80,81]
Both personalized diet and physical activity	5	[58,75,79,82,84]
Health service (doctor or hospital)	8	[66,67,68,69,70,71,72,73]
Personalized motivational messages	7	[46,47,63,64,65,74,83]
Drugs	2	[34,35]
Discussion threads relevant to their condition- and symptom-specific interests	2	[48,49]
Patients with similar health condition	1	[50]

**Table 5 ijerph-19-15115-t005:** The distribution of algorithms in recommendation technology.

Algorithm	Technique	Definition	Number of Studies	Study
Knowledge-based approach	Rule-based approach	Recommendations are generated by using domain knowledge and rules	20	[30,32,33,34,35,37,38,39,41,43,51,52,56,76,78,80,81,83,84,85]
	Case-based reasoning	New recommendations are generated based on past ones	6	[31,53,54,55,58,67]
Collaborative filtering approach	Recommendations are generated by estimating unknown user ratings from similar known user ratings		6	[24,40,47,61,69,73]
Content-based approach	Recommendations are generated based on features from a patient’s past behavior		2	[25,44]
Reinforcement learning approach	Recommendations are adapted based on patients’ feedback		2	[74,79]
Hybrid approach	Multiple recommendation approaches are integrated or assembled		16	[23,27,29,42,45,46,48,50,57,63,65,66,68,70,71,75]
Other approach	New recommendation algorithms designed by researchers		10	[26,32,36,49,59,60,64,72,77,82]

## Data Availability

Data sharing is not applicable to this article as no datasets were generated or analyzed during this scoping review.

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
