# Peer review of "Health Recommender Systems Development, Usage, and Evaluation from 2010 to 2022: A Scoping Review"

_ijerph, 2022, doi:10.3390/ijerph192215115_

Round 1

Reviewer 1 Report

Firstly I would like to thank the editors and the authors for the opportunity to review this interesting paper.

The introduction opening is good and captures the readers attention. The authors provide a strong rationale for the research which is convincing to the reader. However the key terms should be defined as part of setting the scene for the reader. Importantly the authors have not provided a research question. Even though this is a scoping paper this should still be included.

In the methods and materials section the authors include a list of key search engines. These are the common choices for this type of paper, however some justification of their selection backed by references would provide more rigour to the paper. Thes same is true of the inclusion criteria. Some brief justification as to why these are important is needed. I also urge the authors to include papers from 2021 and 2022 to bring the paper up-to-date. The exlusion criteria is self explanatory and clear.  The well known PRISMA approach was applied, this provides the reader with a level of confidence. A brief description of its application is provided, and Figure 1 is also well done and very helpful for the reader. Details of the approach are included in supplement 5. I would like to see details of a brief overview of the approach in section 2.3, rather than in supplement 5, or as well as.

In the results section the descriptions of the 50 papers included are provided in supplement 2. This is well done and provides a good summary of key features. The tables and figures provided in this section are also well done. The analysis of the user model data might be better presented in a table as some of the other results aspects have been.

I feel the discussion section is a bit short, and could be improved by adding a research question in the introduction to provide a structure for the analysis of each section and provide the reader with deeper understanding of the purpose. The authors should also provide a description of the impact or implications of their findings. However the authors do a neat job of drawing on the data from the results section and provide some analysis.

The conclusion provides a brief summary, but should also include the implications of the key points raised.

Reviewer 2 Report

The authors present a thorough and interesting scoping review about health recommender systems. The paper is very concise and written to the point. Some points to improve are given below.

- The paper is perhaps too brief in some parts. I think that the introduction section should be enhanced to give the user more information about the main topic.

- The paper should also discuss the generalizable knowledge about the health recommender systems. They are proposing that additional health domains must be explored but is this necessary to gain generalizable knowledge about developing recommender systems? Why are the explored health conditions not enough?

- The paper does not describe what kind of knowledge has not been generated by the existing research? Is developing user models the main goal of recommender systems?

- Table 4 - the authors should also discuss why the limitations of the algorithms?

- The authors should discuss the concept of dynamic user modeling in more detail. What is it? Why is it important to explore this further? How is it different from obtaining additional data from the user in the form of questionnaires. 

- The authors need to discuss in detail what kinds of evaluations have been performed on health recommendation systems.

Round 2

Reviewer 2 Report

the authors have made required changes to the manuscript.